# Charging modulation of the pyridine nitrogen of covalent organic frameworks for promoting oxygen reduction reaction

Xiubei Yang [1,2], Qizheng An[3], Xuewen Li[1,2], Yubin Fu [4] ✉, Shuai Yang [1], Minghao Liu[1], Qing Xu [1,2] ✉ & Gaofeng Zeng [1,2] ✉

Covalent organic frameworks (COFs) are ideal templates for constructing metal-free catalysts for the oxygen reduction reaction due to their highly tuneable skeletons and controllable porous channels. However, the development of highly active sites within COFs remains challenging due to their limited electron-transfer capabilities and weak binding affinities for reaction intermediates. Herein, we constructed highly active catalytic centres by modulating the electronic states of the pyridine nitrogen atoms incorporated into the frameworks of COFs. By incorporating different pyridine units (such as pyridine, ionic pyridine, and ionic imidazole units), we tuned various properties including dipole moments, reductive ability, hydrophilicity, and binding affinities towards reaction intermediates. Notably, the ionic imidazole COF (*im*-PY-BPY-COF) exhibited greater activity than the neutral COF (PY-BPY-COF) and ionic pyridine COF (*ion*-PY-BPY-COF). Specifically, *im*-PY-BPY-COF demonstrated a half-wave potential of 0.80 V in 0.1 M KOH, outperforming other metal-free COFs. Theoretical calculations and in situ synchrotron radiation Fourier transform infrared spectroscopy confirmed that the carbon atoms in the ionic imidazole rings improved the activity by facilitating binding of the intermediate OOH* and promoting the desorption of OH*. This study provides new insights into the design of highly active metal-like COF catalysts.

The oxygen reduction reaction (ORR) plays critical roles as the essential half-reactions of metal-air batteries and fuel cells[1–5]. In the pursuit of high-efficiency catalysts, metal-free materials have emerged as promising candidates because of their cost-effectiveness and superior chemical stabilities[6–10]. Doping heteroatoms into porous carbon materials has proven to be an effective strategy for improving catalytic performance[11–13]. However, precise control of the location, content, and type of doping sites remains challenging, underscoring the necessity of understanding the relationships between these structures and their corresponding properties.

Covalent organic frameworks (COFs) are porous polymers with well-defined structures, maintained by covalently bonded building blocks[14–20]. The tunable skeletons and porosities of COFs make them useful for various applications, such as gas capture[21–23], molecular separations[24–26], photocatalysis[27–29], and ion sensing or conduction[30–34]. Additionally, COFs constructed with electroactive units have shown promise in electrocatalysis and electronic chemical storage

[1]CAS Key Laboratory of Low-Carbon Conversion Science and Engineering, Shanghai Advanced Research Institute (SARI), Chinese Academy of Sciences (CAS) Shanghai, Shanghai 201210, P. R. China. [2]School of Chemical Engineering, University of Chinese Academy of Sciences Beijing, Beijing 100049, P. R. China. [3]National Synchrotron Radiation Laboratory, University of Science and Technology of China Hefei, Hefei 230029, P.R. China. [4]Center for Advancing Electronics Dresden (cfaed) & Faculty of Chemistry and Food Chemistry, Technische Universität Dresden Dresden, Dresden 01062, Germany. ✉e-mail: yubin.fu@tu-dresden.de; xuqing@sari.ac.cn; zenggf@sari.ac.cn

systems[35–39]. COFs serve as bridges between molecular catalysts and carbon catalysts, offering opportunities to modulate the locations, coordination environment, and densities of active sites[40]. Recently, sulphur-containing COFs were seen to catalyse the ORR[41]. In addition to heteroatom incorporation, the introducing of functional groups to modulate the catalytic performance of the carbon atoms in imine linkage (C = N) has emerged as another important strategy[42–45]. To achieve highly active catalysts, COFs with different functional groups, linkages, and topologies have been synthesized[46–50]. However, their activities still fell short compared to those of carbon catalysts and commercial Pt/C catalysts, and the current record for a half-wave potential is 0.75 V vs. RHE. This limited activity was attributed to the poor electron conductivity of the frameworks and the weak binding of $O_2$ and other oxygen-containing intermediates[51]. Thus, developing electronic conductive frameworks with strong polar sites holds promise for constructing highly active metal-free ORR catalysts[52–57].

In this study, we sought to enhance the catalytic activities of COFs in the ORR by modulating their intrinsic catalytic properties and improving their electronic conductivities. To achieve this goal, we carefully controlled the states of the nitrogen atoms in the COFs, which allowed us to fine-tune various properties such as the electronic conductivity, dipole moments, reductive ability, hydrophilicity, and binding ability of the oxygen-containing intermediate. The *im*-PY-BPY-COF showed a high electronic conductivity of $6.8 \times 10^{-8}$ S cm$^{-1}$, which was 3600 and 440 times greater than those of PY-BPY-COF and *ion*-PY-BPY-COF, respectively. The *im*-PY-BPY-COF demonstrated better ORR activity than other metal-free COFs.

## Results and Discussion
### Chemistry and structure of catalytic COFs

We synthesized the base PY-BPY-COF by reacting **4,4′,4″,4‴-(pyrene-1,3,6,8-tetrayl)-tetraaniline (PY)** with **[2,2′-bipyridine]-5,5′-dicarbaldehyde (BPY)** in a mixture of *n*-butanol/**1,2**-dichlorobenzene (0.5 mL/0.5 mL) containing HOAc (6 M, 0.1 mL). The reaction was conducted at 120 °C for 72 h via a solvothermal method, resulting in a yield of 86% (Supplementary Fig. 1a). Subsequently, the COF was reacted with bromoethane in a postsynthetic strategy to obtain *ion*-PY-BPY-COF (Supplementary Fig. 1b)[58–63]. Furthermore, we synthesized a new COF (*im*-PY-BPY-COF) via a cycloaddition reaction between PY-BPY-COF and paraformaldehyde (Fig. 1a & 1b and Supplementary Fig. 1c).

The skeletons and catalytic centres of the COFs were first investigated with Fourier transform infrared (FT-IR) spectroscopy. The C = N linkages in PY-BPY-COF generated a peak at 1608 cm$^{-1}$ (Supplementary Fig. 2, black curve), which also observed in spectra of the *ion*-PY-BPY-COF and *im*-PY-BPY-COF samples, suggesting preservation of the COF skeletons after ionization of the pyridine units. Additionally, the new bonds at 2928 and 2911 cm$^{-1}$ in the spectrum of *ion*-PY-BPY-COF originated from the ethyl groups attached to the pyridine N sites (Supplementary Fig. 2, green curve). For *im*-PY-BPY-COF, the intensity of the C = N peak was greater than those in the other samples, which was attributed to the formation of new C = N bonds in the imidazole units (Supplementary Fig. 2, red curve). X-ray absorption near-edge spectroscopy (XANES) analysis of the nitrogen K-edge was used to investigate the conversion of pyridine-N to imidazole-N (Supplementary Fig. 3). The c peaks (graphitic N, 408.4 eV) of PY-BPY-COF and *im*-PY-BPY-COF were very close, while the peak for *im*-PY-BPY-COF (C = N-C, 399.9 eV) was significantly weaker than that of PY-BPY-COF, and the b peak (C-N-C, 401.8 eV) of *im*-PY-BPY-COF was stronger than that of PY-BPY-COF. These results indicated the formation of imidazole-N[64,65].

The PXRD patterns of PY-BPY-COF showed peaks at 3.11°, 4.45°, 6.22°, 9.31°, 12.5°, and 24.9° corresponding to the (100), (220), (330), (440), and (001) facets, respectively (Fig. 1c). To refine the crystal structure, Pawley refinements was carried out by self-consistent charge density functional tight-binding method. The theoretical and experimental results showed minor deviations, and the goodness of fit

factors $R_{wp}$ and $R_p$ were 3.16 and 3.56%, respectively. Two stacking models, AA and AB, were used to simulate for the COF, and the PXRD pattern obtained with the AA stacking model agreed well with the experimental results (Supplementary Fig. 4 and Supplementary Data 1). Upon ionization of the pyridine units within the COFs, the crystallinity was well maintained, as evidenced by the intense peak at 3.15° in the PXRD pattern, along with multiple peaks at 6.31°, 9.50°, 13.1° and 24.2° originating from the (110), (220), (330), (440) and (001) facets, respectively (Fig. 1d). The different stacking models used for the simulations confirmed that the AA stacking model was adopted by the *ion*-PY-BPY-COF (Supplementary Fig. 5 and Supplementary Table 1). Similarly, *im*-PY-BPY-COF also exhibited good crystallinity after the imidazole ring was constructed from bipyridine units. The PXRD pattern of *im*-PY-BPY-COF showed peak at 3.26° for the (100) facet, corresponding to a positive shift relative to that of PY-BPY-COF. Peaks originating from the (220), (330), (440), and (001) facets were observed at 6.58°, 9.92°, 13.1°, and 24.1°, respectively (Fig. 1e). Pawley refinements showed that the theoretical values were consistent with the experimental results ($R_{wp} = 2.09\%$ and $R_p = 1.53\%$). The simulated stacking models also confirmed that the COFs adopted AA stacking models rather than AB stacking models (Supplementary Fig. 6 and Supplementary Table 2).

The porosities of the COFs played key roles in mass transport during the catalytic processes. Nitrogen ($N_2$) sorption measurements were conducted on the three COF precursors at 77 K. The basal COF, PY-BPY-COF, exhibited a high Brunauer-Emmett-Teller (BET) surface area of 2258 m$^2$ g$^{-1}$ (Fig. 2a). The pore size distribution revealed a total pore volume of 1.5 cm$^3$ g$^{-1}$ and an average pore size of 2.4 nm for PY-BPY-COF (Fig. 2b). After postfunctionalization of the COFs, the corresponding surface areas and pore volumes had decreased. Compared to those of the original PY-BPY-COF, the BET surface areas of *ion*-PY-BPY-COF and *im*-PY-BPY-COF were lower, at 1820 and 1024 m$^2$ g$^{-1}$, respectively (Fig. 2d & 2g). The decreased BET surface area and pore volume of *im*-PY-BPY-COF were attributed to stacking disorder caused by ionic repulsions, as well as pore filling by trifluoroacetate counterions[57]. The pore sizes of *ion*-PY-BPY-COF and *im*-PY-BPY-COF remained the same as that of PY-BPY-COF, with reduced pore volumes of 1.1 and 0.7 cm$^3$ g$^{-1}$, respectively (Fig. 2e & 2h). The porosities and crystallinities of the COFs were examined with high-resolution transmission electron microscopy (HR-TEM) (Fig. 2c & 2f & 2i). Additionally, the COFs were measured using selected area electron diffraction (SAED). The clear diffraction rings were observed from the SAED patterns, which were assigned to COF crystal planes of (100), (220), and (330), respectively (Supplementary Fig. 7). This is consistent with the FFT and PXRD observations of the COFs, confirming the well-maintained crystalline structures upon the modifications[66].

Solid-state $^{13}$C nuclear magnetic resonance (NMR) spectroscopy was used to examine the carbon arrangements within the three COFs. The $^{13}$C NMR spectrum of PY-BPY-COF showed a peak at 156.1 ppm attributed to the C atoms in C = N, while the peaks ranging from 156.2 to 149.2 ppm were attributed to the carbon atoms in the pyridine rings (Supplementary Fig. 8). The carbon atoms in the PY units showed resonances between 131.1 and 121.3 ppm. With the introduction of ethyl units into the COF, additional peaks appeared at 21.8, 35.2 and 38.3 ppm for *ion*-PY-BPY-COF (Supplementary Fig. 9). The $^{13}$C NMR spectrum of *im*-PY-BPY-COF showed a well-retained C = N peak, with a new peak at 136.3 ppm arising from the formation of imidazole rings. Importantly, the peak corresponding to the carbon atoms adjacent to pyridine nitrogen was shifted from 149.2 to 126.4 ppm, and the peak at 161.3 ppm was attributed to the anions (CF$_3$COO$^-$) in the COFs (Supplementary Fig. 10). These observations confirmed the syntheses of the intended COFs.

The XPS survey spectra confirmed the presence of all the elements in the three COFs (Supplementary Fig. 11). In the high-resolution N 1 s spectrum of PY-BPY-COF, peaks at 398.8 and 399.4 eV were

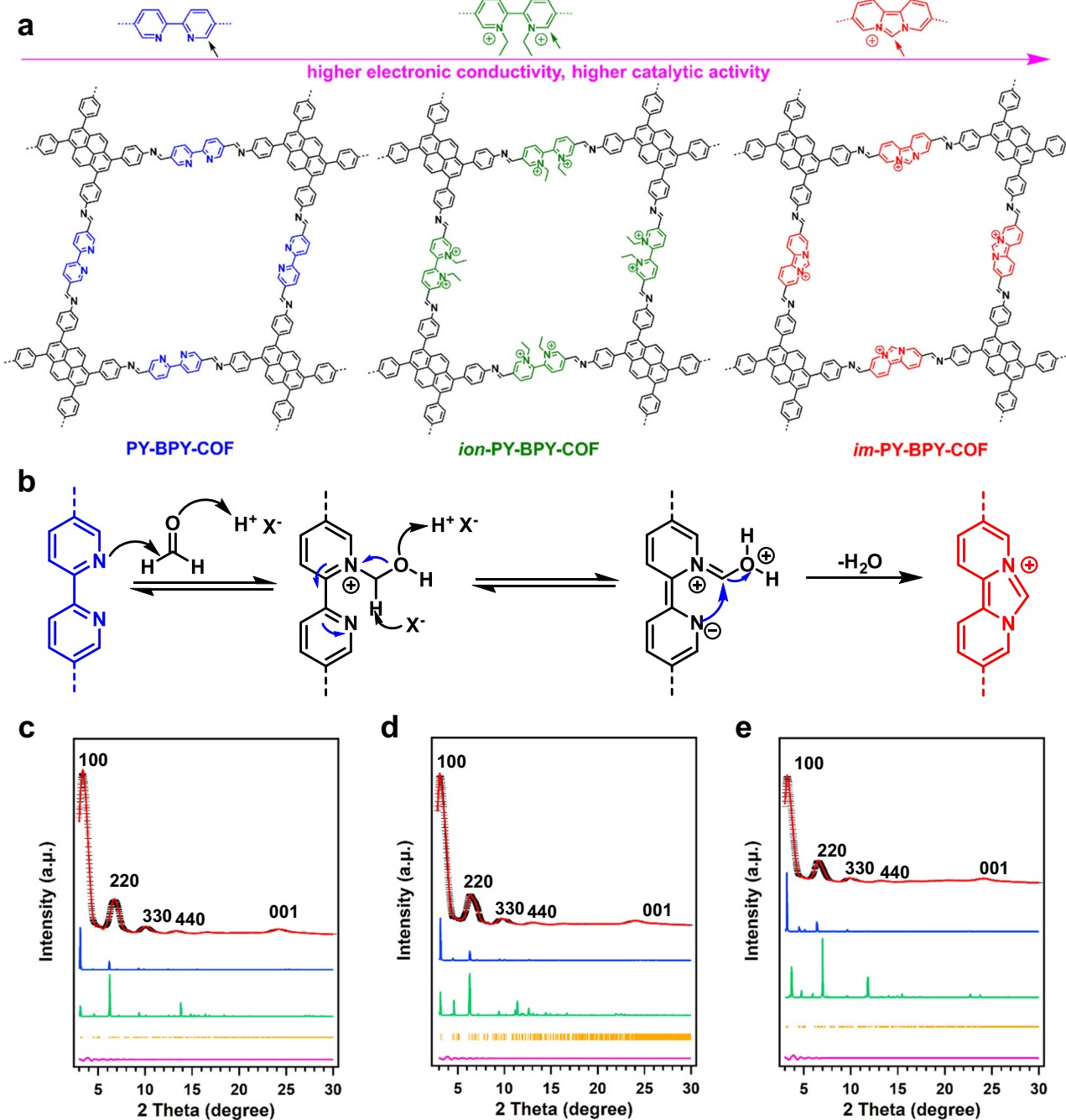

**Fig. 1 | Synthesis and structure of COFs. a** Chemical structure of PY-BPY-COF, *ions*-PY-BPY-COF, and *im*-PY-BPY-COF; **b** Proposed mechanisms for the imidazopyridine bond formation from the BPY units; PXRD patterns for **c** PY-BPY-COF; **d** *ion*-PY-BPY- COF; and **e** *im*-PY-BPY-COF, the experimentally observed (black), Pawley refined (red) and their difference (pink), simulated using the AA (blue), staggered AB (green) stacking modes and Bragger position (orange).

observed, corresponding to pyridine and imine, respectively (Fig. 3a). Upon ionization of the pyridine units, the N 1 s peak disappeared and was replaced by two new peaks at 398.9 and 401.2 eV; these were attributed to the pyridine radical cation and positively charged nitrogen, respectively, confirming the presence of pyridinium in the *ion*-PY-BPY-COF. Similarly, upon protonation of the pyridine units, the N 1 s peak vanished and was replaced by two new peaks at 398.4 and 401.2 eV, indicating the presence of pyridinium in *im*-PY-BPY-COF[63,67,68].

The morphologies of the COFs were examined with scanning electron microscopy (SEM) and transmission electron microscopy (TEM). The SEM images revealed dendritic structures with uniform sizes for PY-BPY-COF, which were consistent with those of the *ion*-PY-

BPY-COF and *im*-PY-BPY-COF. (Supplementary Fig. 12). Moreover, clear layered structures were observed in the TEM images (Supplementary Figs. 13-15). The energy-dispersive X-ray map showed uniform distributions of all the elements within the skeletons. (Supplementary Figs. 16-18). In addition, the C, N, H elemental analysis results are presented in Supplementary Table 3 and agreed with the theoretically calculated values based on the networks.

The electronic conductivities of the COFs were tested with the four-probe method at 298 K. The electronic conductivity of PY-BPY-COF was $1.8 \times 10^{-11}$ S cm$^{-1}$ (Fig. 3b). Upon ionization of the pyridine units, a significant improvement in the electronic conductivity was observed. The *ion*-PY-BPY-COF demonstrated an electronic conductivity of $1.4 \times 10^{-10}$ S cm$^{-1}$ (Fig. 3b). Importantly, *im*-PY-BPY-COF

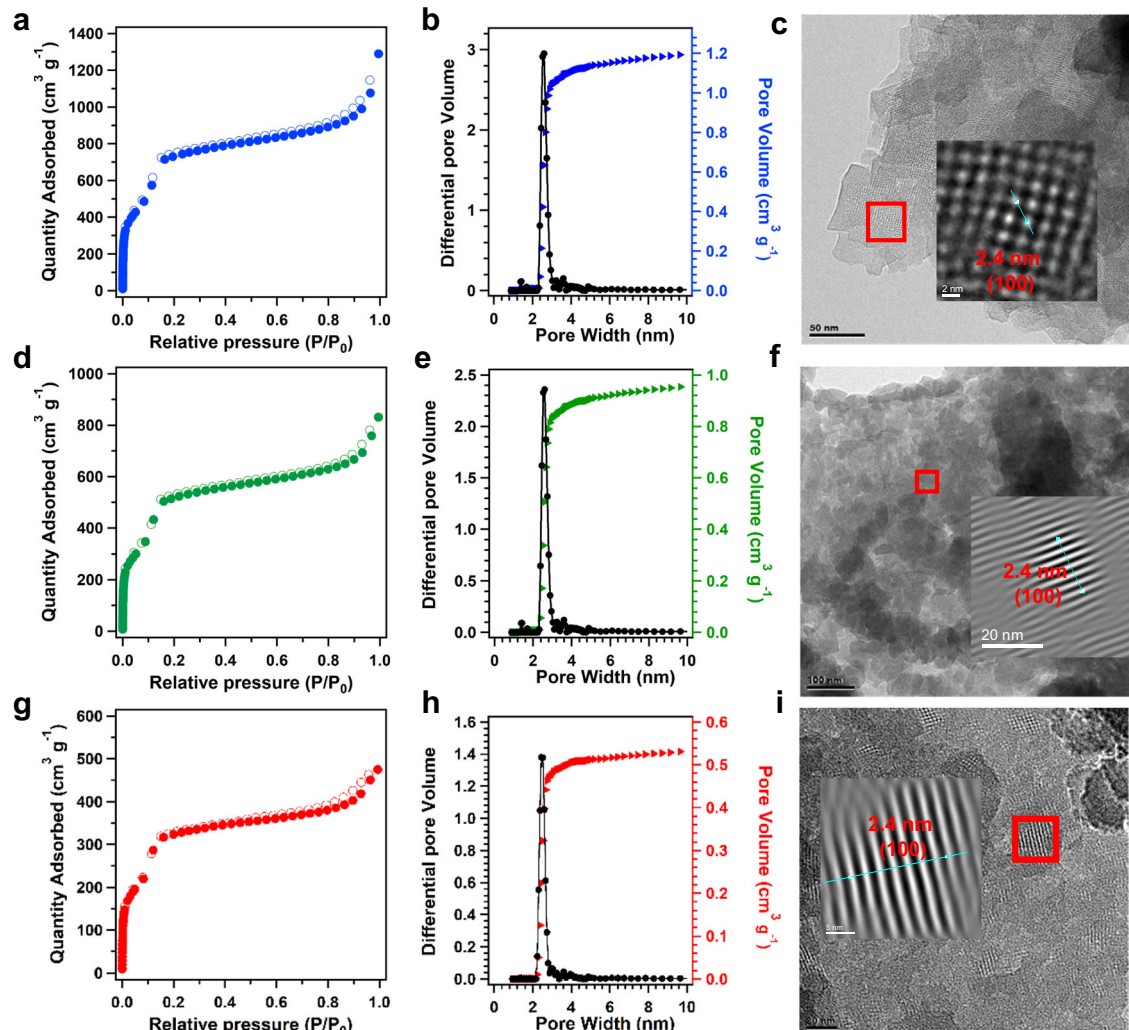

**Fig. 2 | Pore properties of COFs. a** N$_2$ adsorption (solid) and desorption (hollow) profiles at 77 K; **b** Pore size distribution curves; **c** HR-TEM image for PY-BPY-COF; **d**, N$_2$ adsorption (solid) and desorption (hollow) profiles at 77 K; **e** pore size distribution curves, **f** HR-TEM image of *ion*-PY-BPY-COF; **g** N$_2$ adsorption (solid) and desorption (hollow) profiles at 77 K; **h** Pore size distribution curves, **i** HR-TEM image for *im*-PY-BPY-COF.

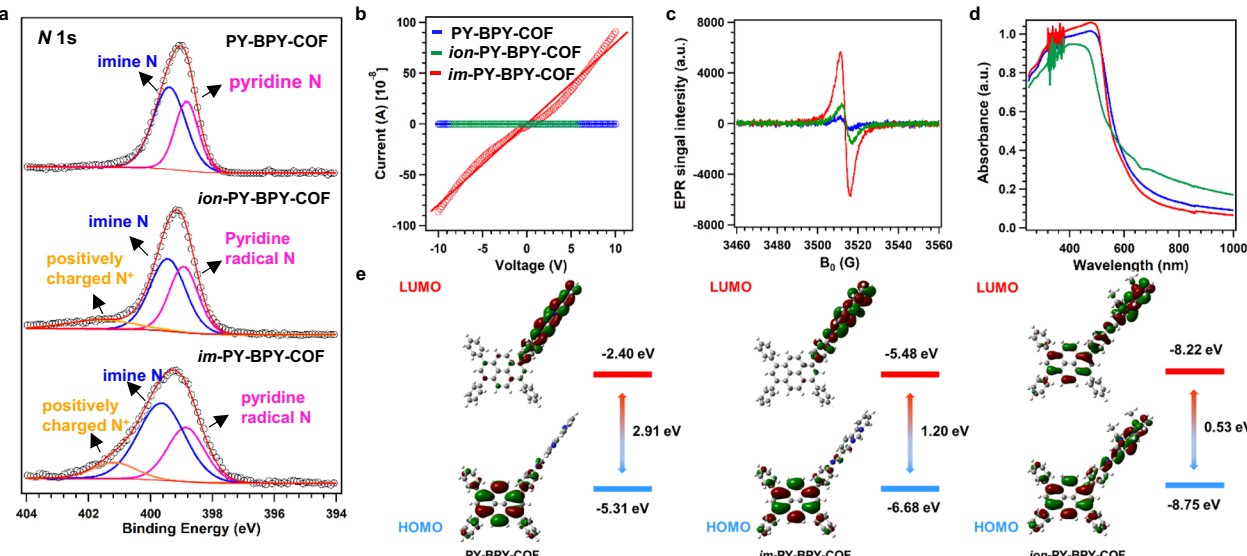

**Fig. 3 | Photoelectric properties of COFs. a** High-resolution N 1 s spectra, **b** Electronic conductivities, **c** EPR spectra under irradiation, **d** Solid-state UV-vis absorption spectra and **e** Molecular orbitals and their energy diagrams calculated at the B3LYP/Def2-SVP level of PY-BPY-COF, *ion*-PY-BPY-COF, and *im*-PY-BPY-COF.

achieved an even higher conductivity of $6.8 \times 10^{-8}$ S cm$^{-1}$, which was 3614 and 441 times greater than that those of unmodified PY-BPY-COF and *ion*-PY-BPY-COF, respectively. The electron paramagnetic resonance (EPR) spectra of the synthesized COF contained a weak peak at 3524 Gs for PY-BPY-COF (Fig. 3c), indicating the presence of organic radicals. The intensity of thew radical peak was noticeably higher in the case of *im*-PY-BPY-COF, indicating a greater contribution of radicals from the ionic imidazole units, resulting in enhanced electron conductivity[68]. The conjugation properties of PY-BPY-COF, *ion*-PY-BPY-COF, and *im*-PY-BPY-COF were studied with solid-state ultraviolet–visible (UV–Vis) absorption spectroscopy (Fig. 3d). By analyzing the Tauc plots derived from the UV–vis absorption spectra, the optical bandgaps were calculated as 1.98, 1.91, and 1.76 eV for PY-BPY-COF, *ion*-PY-BPY-COF, and *im*-PY-BPY-COF, respectively (Supplementary Fig. 19). The narrow bandgap of *im*-PY-BPY-COF favour electronic transport. Additionally, the valence band-XPS (VB-XPS) measurements of the valence band positions for PY-BPY-COF, *ion*-PY-BPY-COF, and *im*-PY-BPY-COF were yielded values of 1.33, 1.37, and 1.38 eV, respectively (Supplementary Fig. 20).

To gain a deeper understanding of the electronic structures of PY-BPY-COF, *ion*-PY-BPY-COF, and *im*-PY-BPY-COF, density functional theory calculations were performed at the B3LYP/Def2-SVP level. The three COFs displayed typical donor-acceptor configurations, with the highest occupied molecular orbitals (HOMO) located mainly at the PY units of the COFs, and the lowest unoccupied molecular orbitals (LUMO) were fully localized over the linkers (Fig. 3e). The LUMO and HOMO energy-levels for PY-BPY-COF were −5.31 and −2.40 eV, respectively, resulting in an energy gap of 2.91 eV. In comparison, the *ion*-PY-BPY-COF exhibited a lower HOMO energy (−8.75 eV) and considerably lower LUMO energy (−8.22 eV), while the *im*-PY-BPY-COF exhibited a lower HOMO energy (−6.68 eV) and substantially lower LUMO energy (−5.48 eV). These findings indicated that the *ion*-PY-BPY-COF had greater reducing ability due to its lower HOMO and LUMO energies. Furthermore, the dipole moments for PY-BPY-COF, *ion*-PY-BPY-COF, and *im*-PY-BPY-COF were 1.24, 43.40 and 39.63 Debye, respectively. The high dipole moments indicated that the ionic imidazole units influenced the intermolecular polarity and promote charge-redistribution within the molecular backbones (Supplementary Fig. 21). This internal charge-transfer that may facilitate the absorption of O-containing intermediates[69]

.The hydrophilicities of the COFs played crucial roles in determining their electrocatalytic ORR activities. The water contact angles for the surfaces of PY-BPY-COF, *ion*-PY-BPY-COF and *im*-PY-BPY-COF were 92°, 75° and 72°, respectively, indicating that the presence of ionic linkers increased the hydrophilicities of the COFs (Supplementary Fig. 22).

The thermal and chemical stabilities of the COFs were investigated. Thermogravimetric analyses (TG) revealed that the three COFs were thermally stable below 400 °C under a N$_2$ atmosphere (Supplementary Fig. 23). To assess their chemical stabilities, the COFs were treated with 0.1 M KOH for one week. PXRD patterns confirmed that the crystallinities of the COFs remained intact after the treatment (Supplementary Fig. 24). Furthermore, the FT-IR spectra exhibited all the expected peaks, indicating preservation of their framework structures (Supplementary Fig. 25).

Thus, while *im*-PY-BPY-COF shares similar crystal and pore structures with PY-BPY-COF and *ion*-PY-BPY-COF, it exhibited a significantly higher electronic conductivity, improved charge transfer ability, enhanced reductive ability, and greater hydrophilicity than the other two COFs.

## ORR performance on catalytic COFs

We investigated the catalytic performance of the COFs with a three-electrode system under alkaline conditions. Cyclic voltammetry (CV) measurements were initially performed at a scan rate of 50 mV s$^{-1}$ with a 0.1 M KOH aqueous solution saturated with N$_2$ or O$_2$ (Supplementary Fig. 26). In an oxygen-saturated environment, *im*-PY-BPY-COF exhibited a clear peak at 0.7 V vs. RHE, indicating its reactivity in the ORR. No peak was observed under nitrogen-saturated conditions. In comparison, the spectra of PY-BPY-COF and *ion*-PY-BPY-COF exhibited reduction peaks at 0.65 and 0.68 V, respectively, these values were 50 and 20 mV more negative than that of *im*-PY-PY-COF, indicating weaker activity. Linear sweep voltammograms (LSV) were obtained with a rotating-disk electrode and a scan rate of 1600 rpm. Initially, the catalytic behaviour of commercial Pt/C was studied as a control. The onset potential (E$_O$) for Pt/C was 0.97 V, corresponding to a current density of 0.1 mA cm$^{-2}$, and the half-wave potential (E$_{1/2}$) was 0.85 V with a limited current density (j$_{lim}$) of 5.6 mA cm$^{-2}$ (Fig. 4a). PY-BPY-COF had E$_O$ and E$_{1/2}$ values of 0.84 and 0.73 V, respectively, with a j$_{lim}$ of 4.2 mA cm$^{-2}$, indicating inferior performance compared to Pt/C. On the other hand, *ion*-PY-BPY-COF demonstrated improved catalytic activity, with an E$_O$ of 0.88 V, E$_{1/2}$ of 0.77 V, and j$_{lim}$ of 5.7 mA cm$^{-2}$. Notably, *im*-PY-BPY-COF exhibited even better catalytic performance, with an E$_O$ of 0.90 V, an E$_{1/2}$ of 0.80 V and a j$_{lim}$ of 5.8 mA cm$^{-2}$. The ORR activity of *im*-PY-BPY-COF was higher than those of most reported metal-free COF catalysts (Supplementary Table 4). The kinetic behaviors were explored by comparing the Tafel slopes. The Tafel slope for *im*-PY-BPY-COF was 56.2 mV dec$^{-1}$, which was markedly lower than those of PY-BPY-COF (79.4 mV dec$^{-1}$), *ion*-PY-BPY-COF (76.8 mV dec$^{-1}$), and Pt/C (117.7 mV dec$^{-1}$), indicating favorable kinetic behavior (Fig. 4b).

To elucidate the distinct catalytic activities of the linkers in COFs, we synthesized bipyridine (BPY), ion-bipyridine (*ion*-BPY) and im-bipyridine (*im*-BPY) compounds (Supplementary Fig. 27). Subsequently, their catalytic activities were investigated (Supplementary Fig. 28). The LSV curves revealed that *im*-BPY had greater activity with an E$_O$ and an E$_{1/2}$ of 0.88 V and 0.72 V, respectively, which were higher than those of BPY and *ion*-BPY. This confirmed that the catalytic activities of the small molecules followed the same trend observed for the COFs, further supporting the contributions of the different linkers. To eliminate the potential difference in charge transfer between carbon nanotubes (CNTs) and small molecules during the catalytic process, we linked small molecules to aminated CNTs. The LSV curves revealed that the catalytic activities of the small molecules attached to CNTs were similar to those of the COFs, further confirming the contributions of the different linkers (Supplementary Fig. 29). In addition, we investigated the ORR activity originating from the CNTs alone. The E$_O$, E$_{1/2}$, and j$_{lim}$ values for the CNTs were 0.85 V, 0.69 V and 3.24 mA cm$^{-2}$, respectively, indicating that the observed activity was indeed originated from the COFs (Supplementary Fig. 30).

In order to further investigate the catalytic activity, we investigated the TOF and mass activity of COF. (Fig. 4c). The *im*-PY-BPY-COF catalyst exhibited the highest TOF of 0.0198 s$^{-1}$ at an experimental potential of 0.7 V, which was 5.4 and 3.2 times greater than the values exhibited by PY-BPY-COF (0.0037 s$^{-1}$) and *ion*-PY-BPY-COF (0.0061 s$^{-1}$), respectively. The corresponding mass activity for *im*-PY-BPY-COF was 0.57 A mg$^{-1}$, which was higher than those of PY-BPY-COF (0.17 A mg$^{-1}$) and *ion*-PY-BPY-COF (0.19 A mg$^{-1}$).

To investigate the catalytic behaviors of the prepared COFs, we measured their electrochemically active surface areas (ECSA). Electrochemical double-layer capacitances (C$_{dl}$) were calculated from CV plots taken over the range 0.92–1.02 V at scan rates of 10–50 mV s$^{-1}$ (Supplementary Fig. 31). The C$_{dl}$ values for *im*-PY-BPY-COF, *ion*-PY-BPY-COF, and PY-BPY-COF were 8.2, 6.8 and 5.8 mF cm$^{-2}$, respectively (Fig. 4d), with corresponding normalized ECSA of 205, 170 and 145, respectively. The larger ECSA indicated that the *im*-PY-BPY-COF contained more exposed active sites during the catalytic process than did the other COFs. The normalized LSV curve for *im*-PY-BPY-COF displayed a higher current density than those of PY-BPY-COF and *ion*-PY-BPY-COF from 0.2 to 0.9 V (Fig. 4e, red curve). This indicated that the excellent catalytic performance

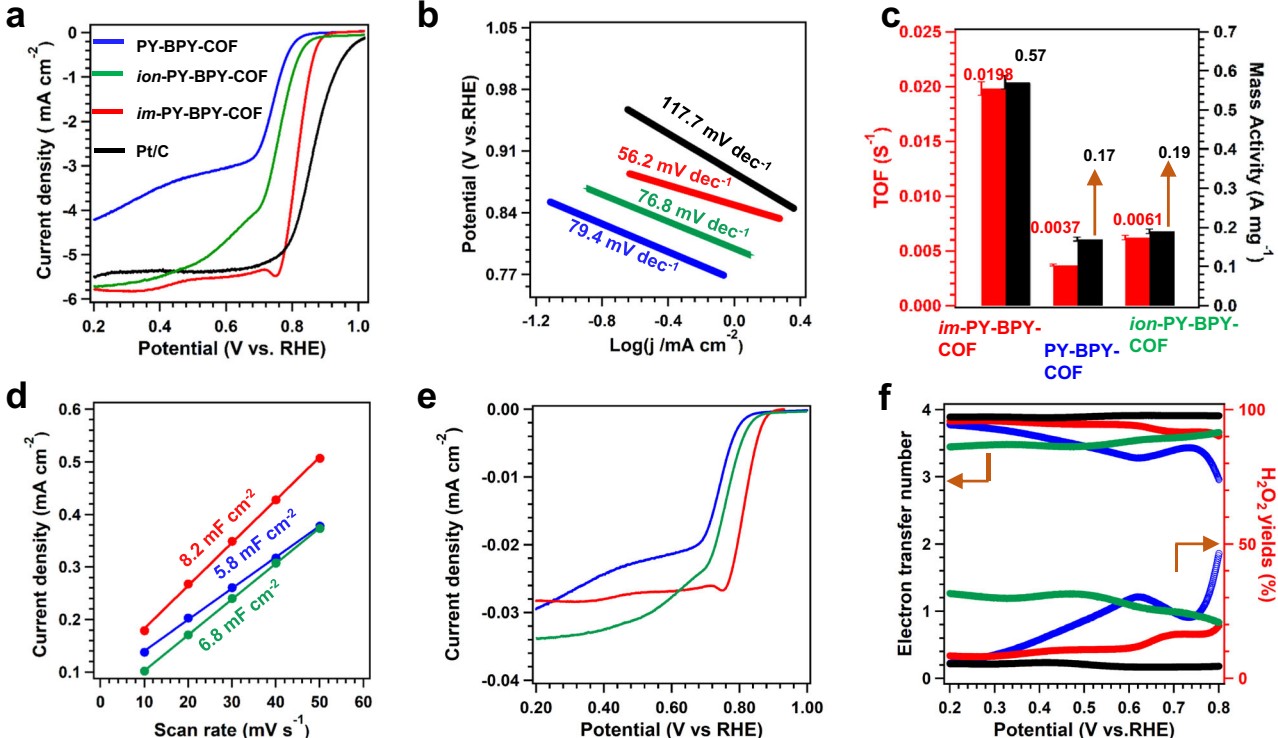

**Fig. 4 | Electrochemical performance of COFs. a** LSV curves; **b** corresponding Tafel slopes; **c** comparisons of the TOF and mass activities; **d** the $C_{dl}$; **e** the normalized LSV curves by the ECSA; **f** electron transfer number and $H_2O$ yield plots determined from the RRDE measurements for Pt/C (black), PY-BPY-COF (blue), *ion*-PY-BPY-COF (green), and *im*-PY-BPY-COF (red).

was primarily due to the intrinsic activity of the carbon sites rather than the ECSA.

The selectivity of the catalyst is another important parameter for the ORR, and was evaluated with a rotating ring-disk electrode. The Pt/C catalyst exhibited an electron transfer number (n) of 3.9–4.0, with hydrogen peroxide ($H_2O_2$) yields of 4% within the potential range of 0.8–0.2 V (Fig. 4f, black curve). In comparison, the *im*-PY-BPY-COF catalyst had n values ranging from 3.6–4.0 and $H_2O_2$ yields of 10%–18% (Fig. 4f, red curve), indicating a $4e^-$ pathway for the ORR. However, PY-BPY-COF and *ion*-PY-BPY-COF exhibited lower selectivity. The n values for PY-BPY-COF ranged from 3.0 to 3.8, with the corresponding $H_2O_2$ yields of 12–50% within the same potential range (Fig. 4f, blue curve). The *ion*-PY-BPY-COF delivered n values of 3.4–3.7, and $H_2O_2$ yields of 20–30% (Fig. 4f, green curve).

The durability of *im*-PY-BPY-COF was tested for 20 h at 0.4 V. The resulting current-time chronoamperometric response displayed a minimal decrease in the current density (only 10%) (Supplementary Fig. 32). Moreover, the PXRD patterns and FT-IR spectra revealed that the crystal structures and frameworks were well preserved after the long-term stability test, indicating high stability in a 0.1 M KOH solution (Supplementary Figs. 33 & 34). In addition, *im*-BY-BPY-COF demonstrated good chemical stability when methanol was added to the electrolytes, as the current density exhibited minor changes (Supplementary Fig. 35).

## Density functional theory calculations

To investigate the catalytic mechanism, theoretical calculations were also conducted. The catalytic sites in the COFs were identified as C atoms with positive charges. Bader charge analyses of all atoms in the COFs revealed that the C atoms adjacent to the N sites in the pyridine units of (C1) for PY-BPY-COF and the carbon atoms between the N atoms in the imidazole units of *im*-PY-BPY-COF (C3) specifically served the catalytic sites for the ORR (Fig. 5a and Supplementary Fig. 36). The

free energy changes (ΔG) for the ORR with the catalytic COFs were calculated as U = 1.23 V. The rate-determining step (RDS) for PY-BPY-COF was found to be the formation of OOH* from $O_2$ (Supplementary Fig. 37), with a ΔG of 2.07 eV (Fig. 5b). In contrast, the positively charged COFs had different RDS steps in their catalytic processes, which involved of OH* to $H_2O$. The ΔG value for *ion*-PY-BPY-COF was 0.74 eV (Supplementary Fig. 38), while *im*-PY-BPY-COF delivered a lower ΔG of 0.38 eV (Supplementary Fig. 39), confirming its superior catalytic activity.

To identify the reaction intermediates and understand the catalytic reaction mechanism, in-situ SR-FTIR spectroscopy was employed, which reliably identified key intermediate species during the reaction process (Supplementary Fig. 40). The results obtained for *im*-PY-BPY-COF under different monitoring conditions are shown in Fig. 5c. An obvious absorption band corresponding to the emergence of OOH* (located at 1,050-1,060 cm⁻¹) was observed once the applied potential exceeded 0.9 V[70], indicating that the carbon atoms in *im*-PY-BPY-COF facilitated the formation of OOH*. Notably, a new IR absorption band for C-OH species was observed at 1640–1650 cm⁻¹ at 0.9 V (Fig. 5d)[53], and its intensity increased with increasing applied potential, suggesting that the carbon sites were active sites for the absorption of OH* to promote ORR process. Linear plots of the in-situ SR-FTIR absorption band intensities for the C-OH and OOH* species with respect to the applied voltage are shown in Fig. 5e. Interestingly, the intensity of the C-OH absorption band increased after only 0.9 V and remained basically unchanged after 0.8 V. Meanwhile, the intensity of the absorption band increased sharply before 0.8 V and then gradually decreased. These pieces of evidence strongly confirmed that carbons in *im*-PY-BPY-COF preadsorbed oxygen to form a highly active OOH* structure, which then spontaneously generated the C-OH that served serving as the final product in the subsequent reaction for fast ORR kinetics. For *ion*-PY-BPY-COF, the intensity changes for the OOH* and C-OH vibrational bands were similar to those observed for *im*-PY-BPY-COF,

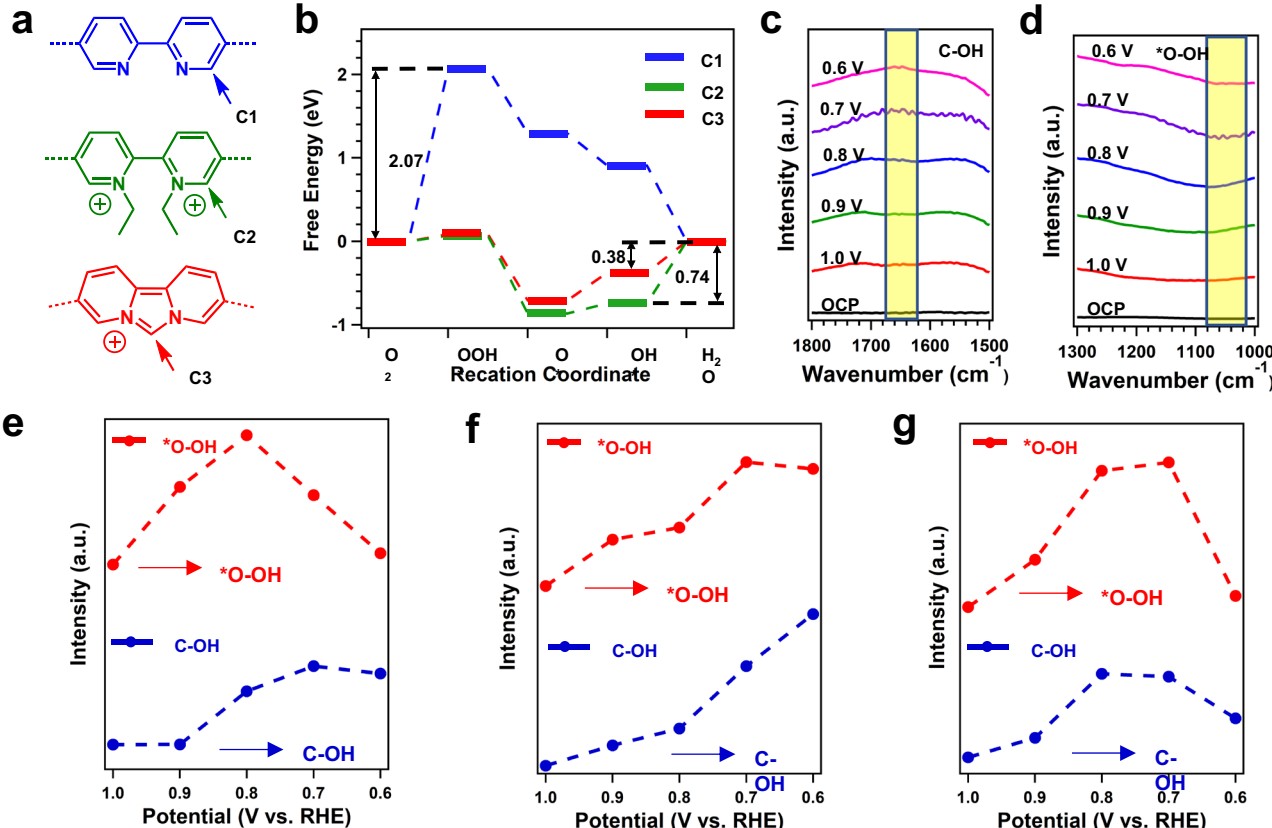

**Fig. 5 | ORR mechanism of COFs. a** Models structures of PY-BPY-COF, *ion*-PY-BPY-COF, and *im*-PY-BPY-COF; **b** Free energy changes for the active carbons under the U = 1.23 V; **c** In-situ SR-FTIR spectroscopy results in the range of 1,300-1,000 cm⁻¹; **d** 1,800-1,500 cm⁻¹ at typical potentials of 0.6 V to 1.0 V for *im*-PY-BPY-COF; intensity difference of the infrared signals at 1,055 cm⁻¹ and 1,644 cm⁻¹ during ORR process of **e** *im*-PY-BPY-COF, **f** *ion*-PY-BPY-COF, and **g** PY-BPY-COF.

indicating that they had the same rate-determining steps in the catalytic process (Fig. 5f and Supplementary Fig. 41). In situ SR-FTIR spectra of PY-BPY-COF revealed that the intensity of the C-OH band followed the same trend as that of OOH*, suggesting that the activity depended solely on the binding ability of OOH*, which was consistent with the theoretical results (Fig. 5g and Supplementary Fig. 42).

## Discussion

In this study, catalytic COFs with different types of pyridine units were developed to catalyse the ORR. The properties of the COFs, such as their electronic conductivities, dipole moments, reductive abilities, and hydrophilicities, were carefully tuned with a precise charge-modifying strategy involving the pyridine units. Moreover, the carbon atoms in the ionic imidazole showed stronger binding affinities for the intermediate OOH* and facilitated the desorption of OH*, thereby enhancing the overall catalytic activity. The optimized COF catalyst showed better performance than other COF-based catalysts. This study provides valuable insights into the design and exploration of ideal active sites in COFs for efficient ORR catalysis.

## Methods

### Synthesis of PY-BPY-COF

The PY-BPY-COF was synthesized according to the previous report[71]. The mixtures of PY (11.3 mg, 0.02 mmol), BPY (8.5 mg, 0.04 mmol), HOAc (6 M, 0.1 mL), *n*-Butanol (0.5 mL) and **1,2**-dichlorobenzene (0.5 mL) of and reacted at 120 °C for 72 h. The obtained solid was washed with THF three times and subjected to Soxhlet extraction with THF. to produce PY-BPY-COF in an isolated yield of 86%.

### Synthesis of *ion*-PY-BPY-COF

The *ion*-PY-BPY-COF was synthesized according to previous report[72]. The PY-BPY-COF (50 mg) was dispersed in a mixture of acetonitrile (4 mL) and bromoethane (1 mL), which was then refluxed under nitrogen protection for 12 h. The product was flittered, washed with different solvents including THF, ethanol and acetone, and then dried under vacuum for 12 h to afford the *ion*-PY-BPY-COF.

### Synthesis of *im*-PY-BPY-COF

The synthesized PY-BPY-COF (18.4 mg, 0.02 mmol) and paraformaldehyde (7.2 mg, 0.24 mmol) were added in the mixed solvents of mesitylene (0.5 mL) and n-BuOH (0.5 mL),. Then trifluoroacetic (0.04 mL) was added into the mixture. The suspension was reacted at 120 °C for 12 h under vacuum. The precipitate was treated by the same methods as that for PY-BPY-COF. with an isolated yield of 63%.

### Synthesis of BPY-CNT

1 g of aminated CNT was mixed with BPY (50 mg) in dioxane solution (100 mL) at 80 °C for 2 h. The production was flitted, washed with THF, and ethanol, and dried under vacuum for 12 h to afford the BPY-CNT.

### Synthesis of ion-BPY-CNT

BPY-CNT (250 mg) was dispersed in a mixture of acetonitrile (10 mL) and bromoethane (2.5 mL) in a flask. The mixture was refluxed for 12 h under nitrogen protection. The production was flitted, washed with THF, and ethanol and dried under vacuum for 12 h to afford the *ion*-BPY-CNT.

## Synthesis of im-BPY-CNT

In a flask, BPY-CNT (250 mg) and 0.3 mL of diiodomethane were dissolved in 10 mL of anhydrous acetonitrile under atmosphere. The reaction mixture was stirred at 100 °C for 48 h. The production was flitted, washed with THF, and ethanol and dried under vacuum for 12 h to afford the *im*-BPY-CNT.

## Electrochemical measurements

The electrochemical measurements were conducted under the same condition according to previous report[73]. Firstly, the homogeneous catalysts ink was prepared by dispersing 5 mg of COFs and 3 mg of CNTs in 500 μL Nafion ethanol solution (0.25 wt.%) through 1 h to sonification treatment. Then, 16 μL prepare ink dropped onto a glassy carbon electrode (diameter 5.00 mm, surface area 0.125 cm$^2$). The electrochemical measurements of COFs catalysts were carried out in a typical three-electrode cell at room temperature over an electrochemical workstation (Pine Research Instrumentation, USA). Platinum wire and silver/silver chloride (Ag/AgCl) electrode (saturated with 3 M KCl) were served as counter and reference electrodes, respectively. For evaluating the ORR activity and selectivity of different catalysts, a rotating ring disk electrode (RRDE) consisting of a platinum ring and a glassy carbon disk was used as the working electrode substrate. The RRDE measurements were performed at a rotation rate of 1600 ppm with a sweep rate of 10 mV s$^{-1}$. The ORR electrochemical experiments were conducted in O$_2$-saturated KOH solutions (0.1 M).

## In situ Fourier transform infrared spectroscopy measurements

In situ synchrotron radiation Fourier transform infrared spectroscopy (SR-FTIR) investigations were carried out at the beamline BL01B of the National Synchrotron Radiation Laboratory, China[74]. A customized top-plate cell-reflection infrared reactor, which was equipped with a ZnSe crystal as the infrared transmission window (cutoff energy of ~625 cm$^{-1}$), was used as in situ FTIR reactor (Supplementary Fig. 43). The FTIR spectrometer (Bruker 66 v/s) contains a KBr beam splitter and a liquid-nitrogen-cooled HgCdTe detector. In addition, the SR-FTIR system was also equipped with an infrared microscope (Bruker Hyperion 3000) with a magnification factor 16 objective lens. This configuration allowed for infrared spectroscopy measurements across the wide range of 15 – 4000 cm$^{-1}$ with a resolution of 0.25 cm$^{-1}$. To minimize the loss of infrared light, the catalyst electrode was tightly sealed under the ZnSe crystal window, maintaining a micrometer-level gap. All the infrared spectra were recorded after applying a constant potential to the catalyst electrode for 30 min. To obtain high quality SR-FTIR spectra, a reflection mode with a vertical incidence of infrared light was used, and each spectrum was scanned for ~514 times with a resolution of 2 cm$^{-1}$.

## Data availability

The authors declare that all the data supporting the findings of this study are available within the article. The Supplementary Information, Source Data, and full image dataset are also available from the corresponding author upon request.

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

## Acknowledgements

The authors acknowledge the financial supports from the National Natural Science Foundation of China (52303288, 22075309, 22378413), Science and Technology Innovation Plan of the Science and Technology Commission of Shanghai Municipality (22ZR1470100, 23DZ1202600, 23DZ1201804), the Youth Innovation Promotion Association of Chinese Academy of Sciences (E324441401), and Biomaterials and Regenerative Medicine Institute Cooperative Research Project Shanghai Jiao Tong University School of Medicine (2022LHA09). Dr. Yubin Fu gratefully acknowledge the GWK support for funding this project by providing computing time through the Center for Information Services and HPC (ZIH) at TU Dresden. The authors also thank the infrared beamline BL01B from the National Synchrotron Radiation Laboratory for proving support for materials characterization.

## Author contributions

Q. X. conceived the idea and designed the experiments. X. Y. and Q. A. performed the experiments. Y. Fu contributed to the theory calculation part. X. L., S. Y. and M. L. participated in some experiments. X. Y., Q. X. and G. Z. wrote and revised the manuscript. All the authors contributed to the data interpretation, discussion, and manuscript revision. All authors have given approval to the final version of the manuscript. X. Y., and Q. A. contributed equally to this work.

## Competing interests

The authors declare no competing interests.
