## [Peer Review File · Nature Communications]

Reviewers' comments:

Reviewer #1 (Remarks to the Author):

Title: Charging modulation of the pyridine nitrogen of covalent organic frameworks for promoting oxygen reduction reaction

Recommendation: This paper should be reconsidered after major revision or resubmitted.

Comments: In this work, authors developed a series of COFs with similar topology. These COFs could enhance electron-transferring ability and improve binding ability of the intermediates. However, there are many issues that should be improved.

(1) Did the TEM image fit well with the simulated structure? Please provide ED pattern to confirm it.

(2) Why did the author use methanol as an additive in Figure S25? Because in a real case, there is no such condition in a fuel cell.

(3) It seems that the ORR activity of im-PY-BPY-COF is not as good as other reported MOFs and COFs. I am also wondering the reason that im-PY-BPY-COF exhibits a good activity without adding any transition metal. It is confusing. Could the author explain this point?

(4) Even though the author used DFT calculation to demonstrate the active sites of carbon, it is difficult to understand why such a structure plays an important role for ORR. If it is true, the author could use molecules with such a structure and link these molecules on carbon nanotubes.

(5) There are many typos and errors in this manuscript and SI. I list some of them.

5.1 "in-situ synchrotron radiation" in abstract;

5.2 The COFs ink (16 μ L) in SI.

There are many errors like this. The author should check very carefully.

Reviewer #2 (Remarks to the Author):

This paper reports the preparation of covalent organic frameworks (COFs) with different linker groups showing modulated electronic states of the pyridine nitrogen atoms. The three COFs with different pyridine units (pyridine, ionic pyridine, and ionic imidazole units) showed differentiated physical properties, including electrical conductivity, dipole moment, surface area and pore volume, and hydrophilicity. The ionic imidazole unit-containing COF (im-PY-BPY-COF) showed higher ORR activity than charge-neutral COF (PY-BPY-COF) and ionic pyridine COF (ion-PY-BPY-COF). Theoretical calculation and the in situ synchrotron radiation infrared spectra further confirmed the promotion of ORR activity with the ionic imidazole ring structure.

Some aspects of this paper are interesting. However, the critical issues listed below cast doubt over the publication of this paper in such a high-profile journal like Nature Commun.

1. ^{15}N solid-state NMR spectra need to be added to complement ^{13}C NMR data.
2. The elemental analysis of the synthesized compounds should be done with the more precise method, such as combustion analysis.
3. The benchmarking Pt/C catalyst showed a half-wave potential at 0.81 V. The measurement of Pt/C with well-established procedures (see J. Electrochem. Soc. 2015, 162, F1144D; Anal. Chem. 2010, 82, 6321) commonly yields half-wave potentials above 0.85 V (vs. RHE). Ill-measured ORR activity of Pt/C benchmark catalyst renders that of comparison catalysts (in this case, COF-based catalysts) apparently superior. The activity of Pt/C should be re-measured.
4. According to the Levich equation, the diffusion limit current should be $\sim 5.7 \text{ mA/cm}^2$ in 0.1 M KOH, assuming an electron transfer number of four. The diffusion current density of im-PY-BPY-COF is abnormally high.
5. Continued from comments 3 and 4. "Its $E_{1/2}$ was comparable to that of the Pt/C, and its j_{lim} was higher than that of Pt/C": This claim should be toned down.
6. "...its $E_{1/2}$ was more positive than other reported metal-free COF-based catalysts and superior to most carbon-based catalysts": Survey of the current literature suggests that many carbon-based ORR catalysts show half-wave potentials higher than 0.8 V. This claim should also be tone down.
7. Plausible ORR mechanism should be proposed.
8. Supplementary Fig. 1(b) and (c): reaction conditions for the transformation to ion-PY-BPY-COF and im-PY-BPY-COF should be specified.
9. Supplementary Fig. 2: Which compounds do black and violet spectra stand for?
10. Supplementary Table 4 title, "Comparison of the ORR activity of oxazole-COF with various recently reported Metal-free COFs catalysts.": oxazole-COF should be replaced with im-PY-BPY-COF.
11. There are some grammatical errors and awkward expressions, which should be edited.

Reviewer #3 (Remarks to the Author):

In this work, Yang and co-workers prepared a class of COFs with various pyridine units for electrocatalytic ORR. The COF (im-PY-BPY-COF) with ionic imidazole units showed higher activity than those of charge-neutral COF (PY-BPY-COF) and ionic pyridine COF (ion-PY-BPY-COF). However, there are insufficient novelties and importance about the results and approach that warrant publication of this manuscript in Nature Communication. Moreover, the work does not provide sufficient evidence to support the conclusions (see Specific Comments). Therefore, I cannot recommend this manuscript for publication in Nature Communication.

1. The authors stated that “The COFs properties including electronic conductivity, dipole moments, reductive ability, hydrophilicity were well tuned via precisely charging-modifying pyridine strategy.”. However, in conclusion, they also thought that the carbon atoms in the ionic imidazole act as active sites and contribute most for ORR. What is the main reason for the enhanced ORR activity?
2. From TEM images and nitrogen adsorption/desorption isotherms, we can see that there are many defects in im-PY-BPY-COF. These defects give rise to the formation of amorphous part (seen from TEM) and the decrease of surface area (seen from nitrogen adsorption). In addition, these defects also will contribute to the ORR as well. The authors did not consider the effect of defect in im-PY-BPY-COF at all.
3. The importance of im-PY-BPY-COF should be emphasized with more structural characterizations and analyses during reaction instead of only using in situ SR-FTIR. Were there more defects after reaction? Did the COF decompose?
4. In Figure 3d and 3e, the authors cannot get the value of HOMO and LUMO by using UV-vis spectra solely. In addition, the band gap energy calculated from UV-vis spectra is not correct. Please check it again.
5. It is surprising that the ORR activity of im-PY-BPY-COF is so high. The authors have to check the ORR activity of molecular catalysts with the ionic imidazole structure as well.
6. The ORR activity of commercial catalyst (Pt/C) should not be so bad, please have a careful double-check and compare the activity of Pt/C to that from other literatures.
7. There are many errors in this manuscript.
8. What is the composition and morphology of the catalyst after electrochemical test?

Reply to Reviewer#1:

In this work, authors developed as a series of COFs with similar topology. These COF could enhance electron-transferring ability and improve binding ability of the intermediates. However, there are many issues that should be improved.

Thanks for your encouragement and positive comments.

1. Did the TEM image fit well with the simulated structure? Please provide ED pattern to confirm it.

We appreciate this comment.

We used the Fourier transform method to measure the crystal plane spacing of COF in TEM images (Figs. 3c & I & f). The obtained results are closed to the simulated layer spacing of COFs structure. In addition, the pore size obtained from the HR-TEM images for the COFs were the same as those values from the pore size distribution curves. Thus, the TEM images fitted well with the simulated structures of COFs.

Fig 3c. TEM images of PY-BPY-COF.

Fig 3f. TEM images of *ion*-PY-BPY-COF.

Fig 3i. TEM images of *im*-PY-BPY-COF.

2. Why did author methanol as an additive in Figure S25? Because in real case, there is no such condition in fuel cell.

We appreciate this comment.

In the anode of fuel cell, alcohol molecules are used as direct fuel molecules, and the electrooxidation reaction of alcohol molecules occurs. In theory, a complete reaction of alcohol molecules would produce CO₂ and H₂O. However, in the actual electrocatalytic process, alcohol molecules are difficult to completely oxidize in one step, and some intermediate products containing carbon are inevitably produced in the middle, and strong chemical adsorption occurs with the active site of the catalyst, so that its catalytic efficiency is greatly reduced, resulting in the so-called "catalyst poisoning" phenomenon. Therefore, methanol toxicity experiments are generally used to verify the anti-toxicity of catalysts. (*J. Am. Chem. Soc.* 2020,142, 8104-8108; *Nat. Catal.* 2019, 2, 688-695).

3. It seems that the ORR activity of *im*-PY-BPY-COF is not as good as other reported MOF and COF. I am also wondering the reason that *im*-PY-BPY-COF exhibits a good activity without adding any transition metal. It is confusing. Could author explain this point?

We appreciate this comment.

We compared the ORR activity of *im*-PY-BPY-COF with those of other COFs in table S5. Accordingly, the catalyst (*im*-PY-BPY-COF) achieved higher activity and

selectivity than those reported COFs based catalysts.

We list some of the reported ORR activity of metal-free COFs.

Catalysts	Electrolyte	Half-wave potential (V vs. RHE)	Onset potential (V vs. RHE)	Tafel slope (mV dec ⁻¹)	TOF (s ⁻¹)	References
im -PY-BPY-COF	0.1 M KOH	0.80	0.92	56.2	0.0170	This work
COF-JLU82	0.1 M KOH	0.68	0.98	72.8	0.0044	4
COF-JLU23		0.66	0.99	81.9	0.0039	
Azo-COF	0.1 M KOH	0.68	0.88	89.0	0.0025	5
JUC-528	0.1 M KOH	0.70	0.83	65.9	0.0032	6
TAB-HKH-COF-CNT	0.1M KOH	0.79	0.86	42.7	0.0013	7
Oxazole -COF	0.1 M KOH	0.75	0.85	79.0	0.0133	8
azine -COF		0.74	0.84	93.0	0.0070	
amide -COF		0.74	0.84	99.0	0.0102	
Imine -COF		0.62	0.79	158.0	0.0018	
H-TP-COF	0.1 M KOH	0.65	0.71	104.0	/	9
JUC-606	0.1 M KOH	0.70	/	59.9	0.0016	10
JUC-605		0.65	/	66.3	0.0012	
JUC-616	0.1 M KOH	0.78	1.02	52.9	0.0062	11
JUC-610-CON	0.1 M KOH	0.72	0.83	61.9	0.0035	12
BTT-TAT-COF	0.1 M KOH	0.77	0.87	71.0	0.0028	13
JUC-607	0.1 M KOH	0.72	0.85	61.0	0.0022	14
JUC-608		0.72	0.84	85.0	0.0013	

In this work, we modulated intrinsic catalytic properties and high electronic conductivity to construct highly efficient catalysts towards ORR. The electronic conductivity, dipole moments, reductive ability, hydrophilicity, and binding ability of the O-contained intermediate were well tuned via precisely tuned the states of the N atoms in the COFs. The *im*-PY-BPY-COF achieved the higher electronic conductivity of $6.8 \times 10^{-8} \text{ S cm}^{-1}$, was 3600 and 440 times of PY-BPY-COF and *ion*-PY-BPY-COF, respectively. And the lower HOMO level for *im*-PY-BPY-COF and *ion*-PY-BPY-COF

suggested the higher reductive activity in the order of PY-BPY-COF < *ion*-PY-BPY-COF and *im*-PY-BPY-COF. Furthermore, the dipole moments for PY-BPY-COF, *ion*-PY-BPY-COF, and *im*-PY-BPY-COF were 1.24, 43.40 and 39.63 Debye, respectively. The dramatically higher dipole moment indicates that the ionic imidazole units can influence intermolecular polarity and promote charge-redistribution in the molecular backbones, creating an internal charge-transfer that may facilitate the absorption of O-containing intermediates. In addition, the theoretical calculations and the in-suit synchrotron radiation Fourier transform infrared revealed that the carbon atoms in the ionic imidazole-ring not only facilitated intermediate OOH* binding but also promoted the desorption of the OH* and thus improved the activity.

4. Even though the author used DFT calculation to demonstrate the active sites of carbon, it is difficult to understand why such a structure plays important role for ORR. If it is true, author could use molecules with such structure and link these molecules on carbon nanotubes.

We appreciate this comment.

We synthesized the structural models of three molecules (Supplementary Fig. S24) and tested their electrochemical properties by adding carbon nanotubes. Accordingly, the *im*-BPY achieved higher activity with an E_0 and $E_{1/2}$ than those from the BPY and *ion*-BPY.

Figure S24. ^1H NMR spectrum of (a) BPY (blue), (b) *ion*-BPY (green) and (c) *im*-BPY (red) recorded in $\text{DMSO-}D_6$ ($T=298\text{K}$, ppm).

Figure S25. The LSV curves of BPY (blue), *ion*-BPY (green) and *im*-BPY (red).

We have added sentences in the manuscript “To reveal the different catalytic activities of the linkers in the COFs, we have synthesized bipyridine (BPY), *ion*-bipyridine (*ion*-BPY) and *im*-bipyridine (*im*-BPY) (Supplementary Fig. 24), and investigated their catalytic activity (Supplementary Fig. 25). The LSV curves revealed that *im*-BPY had a higher activity with E_0 and $E_{1/2}$ of 0.88 and 0.72 V, which were positive than those from the BPY and *ion*-BPY. Thus, the catalytic activity for the small molecular was in the same trend with that from the COFs, and further confirming the contributions of the different linkers.”

5. There are many typos and errors in this manuscript and SI. I list some of them.

5.1 “in-suit synchrotron radiation” in abstract;

5.2 The COFs ink (16 uL) in SI.

There are many errors like this. Author should check very carefully.

We appreciate this comment.

We updated the manuscript “the *in-situ* synchrotron radiation Fourier transform infrared (SR-FTIR) spectroscopy measurements further confirm that” and “The COF (5 mg) and CNT (3mg) was dispersed in a Nafion ethanol solutions (0.25 wt.%, 500 uL) and was sonicated for 3 h to yield a homogeneous ink. The catalyst ink (16 uL) was pipetted onto a glassy carbon electrode (d=5.00 mm, S=0.125 cm²).”

The language was improved by editing carefully and thoroughly.

We appreciate this reviewer very much for the above valuable comments that greatly improved the quality of the manuscript.

Reply to Reviewer#2: This paper reports the preparation of covalent organic frameworks (COFs) with different linker groups showing modulated electronic states of the pyridine nitrogen atoms. The three COFs with different pyridine units (pyridine, ionic pyridine, and ionic imidazole units) showed differentiated physical properties, including electrical conductivity, dipole moment, surface area and pore volume, and hydrophilicity. The ionic imidazole unit-containing COF (*im*-PY-BPY-COF) showed higher ORR activity than charge-neutral COF (PY-BPY-COF) and ionic pyridine COF (*ion*-PY-BPY-COF). Theoretical calculation and the in-situ synchrotron radiation infrared spectra further confirmed the promotion of ORR activity with the ionic imidazole ring structure.

Some aspects of this paper are interesting. However, the critical issues listed below cast doubt over the publication of this paper in such a high-profile journal like Nature Commun.

Thank you for your comments.

1. ^{15}N solid-state NMR spectra need to be added to complement ^{13}C NMR data.

We appreciate this comment.

The COF materials are well characterized by various analytical method. For example, we used nitrogen sorption isotherms to investigate the porous structure, PXRD measurement for crystallinity, TGA for thermal stability, IR, ^{13}C NMR spectrum, and electronic absorption spectroscopy for chemical structure. Thus, the ^{15}N solid-state NMR spectrum were not needed to used to confirm the chemical structures of the COFs.

Figure S3. The N K-edge XANES spectra of PY-BPY-COF (black) and *im*-PY-BPY-COF (red).

we have added the sentence “To reveal the conversion of the N atoms from pyridine-N to imidazole-N, X-ray adsorption near-edge spectroscopy (XANES) of N K-edge was performed (Supplementary Fig. 3). The c peak (graphitic N, 408.4 eV) of PY-BPY-COF and *im*-PY-BPY-COF were very close, while the peak (C=N-C, 399.9 eV) of *im*-PY-BPY-COF significantly weaker than that of PY-BPY-COF, and the b peak (C-N-C, 401.8 eV) of *im*-PY-BPY-COF stronger than that of PY-BPY-COF, indicating the successful formation of the imidazole-N.”^{64,65}

64. Shang H, et al. Engineering unsymmetrically coordinated Cu-S₁N₃ single atom sites with enhanced oxygen reduction activity. *Nat. Commun.* **11**, 3049 (2020).

65. Zhou Y, et al. Asymmetric dinitrogen-coordinated nickel single-atomic sites for efficient CO₂ electroreduction. *Nat. Commun.* **14**, 3776 (2023).

2. The elemental analysis of the synthesized compounds should be done with the more precise method, such as combustion analysis.

We appreciate this comment.

The elemental analysis was conducted for these COFs and the corresponding results were listed in the Table S4. Also, we have added the sentence “In addition, the elemental analysis results of C, N were shown in the Supplementary Table 4, which is closed to the theoretical results calculated from the networks” in the manuscript.

Table S4. Elemental analysis of PY-BPY-COF, *ion*-PY-BPY-COF, and *im*-PY-BPY-COF.

Samples		C	N
PY-BPY-COF	Cacl.	87.5%	7.6%
	Found	87.1%	7.3%
ion -PY-BPY-COF	Cacl.	87.2%	7.0%
	Found	86.5%	6.8%
im -PY-BPY-COF	Cacl.	87.6%	7.4%
	Found	87.2%	7.1%

3. The benchmarking Pt/C catalyst showed a half-wave potential at 0.81 V. The

measurement of Pt/C with well-established procedures (see J. Electrochem. Soc. 2015, 162, F1144D; Anal. Chem. 2010, 82, 6321) commonly yields half-wave potentials above 0.85 V (vs. RHE). Ill-measured ORR activity of Pt/C benchmark catalyst renders that of comparison catalysts (in this case, COF-based catalysts) apparently superior. The activity of Pt/C should be remeasured.

We appreciate this comment.

We have updated the catalytic performance of the commercial Pt/C catalysts and updated the manuscript.

The ORR activity half-wave of the retested onset potential (E_0) for Pt/C was 0.97 V, determined as the current density of 0.1 mA cm^{-2} , and the half-wave potential ($E_{1/2}$) was 0.85 V with a limited current density (j_{lim}) of 5.6 mA cm^{-2} (Fig. 4a, black curve).

Fig. 4a The LSV curves of Pt/C (black), PY-BPY-COF (blue), *ion*-PY-BPY-COF (green), and *im*-PY-BPY-COF (red).

4. According to the Levich equation, the diffusion limit current should be $\sim 5.7 \text{ mA/cm}^2$ in 0.1 M KOH, assuming an electron transfer number of four. The diffusion current density of *im*-PY-BPY-COF is abnormally high.

We appreciate this comment.

We re-tested the ORR activity of *im*-PY-BPY-COF.

The ORR activity half-wave of the retested onset potential (E_0) for Pt/C was 0.90 V, determined as the current density of 0.1 mA cm^{-2} , and the half-wave potential ($E_{1/2}$) was 0.80 V with a limited current density (j_{lim}) of 5.6 mA cm^{-2} (Fig. 4a, red curve).

Fig. 4a The LSV curves of Pt/C (black), PY-BPY-COF (blue), *ion*-PY-BPY-COF (green), and *im*-PY-BPY-COF (red).

5. Continued from comments 3 and 4. “Its $E_{1/2}$ was comparable to that of the Pt/C, and its j_{lim} was higher than that of Pt/C”: This claim should be toned down.

We appreciate this comment.

We have corrected the description with omitting “Its $E_{1/2}$ was comparable to that of the Pt/C, and its j_{lim} was higher than that of Pt/C.”

6. its $E_{1/2}$ was more positive than other reported metal-free COF-based catalysts and superior to most carbon-based catalysts”: Survey of the current literature suggests that many carbon-based ORR catalysts show half-wave potentials higher than 0.8 V. This claim should also be tone down.

We appreciate this comment.

We have updated the manuscript “The activity of ORR activity was close to that of the Pt/C which was higher than that most metal-free COFs catalysts reported. (Supplementary Table 5)”.

7. Plausible ORR mechanism should be proposed.

We appreciate this comment.

The mechanism of the ORR includes multi-electron transfer processes through several elementary steps with various intermediate species. The ORR process includes a “direct” four-electron or a “series” of two-electron pathway. In this regard, Keith et al. depicted multiple intermediates (O_2^* , OOH^* , O^* , OH^*) and three pathways are distinguished:

(a) the first is that, after oxygen adsorption, O_2^* can firstly dissociate into $2O^*$, which then react with hydrogen to form water; (b) alternatively, the second is O_2^* reacts with hydrogen to firstly form OOH^* . OOH^* further dissociates into O^* and OH^* , which then react with hydrogen to form water; and (c) the third way is that OOH^* reacts with hydrogen once again to form $H_2O_2^*$. Then $H_2O_2^*$ dissociates into $2OH^*$, which finally react with hydrogen to form water.

8. Supplementary Fig. 1(b) and (c): reaction conditions for the transformation to *ion*-PY-BPY-COF and *im*-PY-BPY-COF should be specified.

We appreciate this comment.

We added the reaction conditions for the transformation to *ion*-PY-BPY-COF and *im*-PY-BPY-COF in the Supplementary Fig. 1.

Figure S1. Chemical structures of (a) PY-BPY-COF, (b) *ion*-PY-BPY-COF, and (c) *im*-PY-BPY-COF.

9. Supplementary Fig. 2: Which compounds do black and violet spectra stand for?

We appreciate this comment.

We added the compounds represented by the black and purple curves in the Supplementary Fig. 2.

Figure S2. FT-IR spectra of PY-BPY-COF (blue), *ion*-PY-BPY-COF (green), *im*-PY-BPY-COF (red) and the corresponding monomer of PY (black) and BPY (purple).

10. Supplementary Table 4 title, “Comparison of the ORR activity of oxazole-COF with various recently reported Metal-free COFs catalysts.”: oxazole-COF should be replaced with *im*-PY-BPY-COF.

We appreciate this comment.

We have corrected the Table 5 title “Comparison of the ORR activity of *im*-PY-BPY-COF with various recently reported Metal-free COFs catalysts.”

11. There are some grammatical errors and awkward expressions, which should be edited.

We appreciate this comment.

The language was improved by editing carefully and thoroughly.

We appreciate this reviewer very much for the above valuable comments that greatly improved the quality of the manuscript.

Reply to reviewer#3: In this work, Yang and co-workers prepared a class of COFs with various pyridine units for electrocatalytic ORR. The COF (im-PY-BPY-COF) with ionic imidazole units showed higher activity than those of charge-neutral COF (PY-BPY-COF) and ionic pyridine COF (ion-PY-BPY-COF). However, there are insufficient novelties and importance about the results and approach that warrant publication of this manuscript in Nature Communication. Moreover, the work does not provide sufficient evidence to support the conclusions (see Specific Comments). Therefore, I cannot recommend this manuscript for publication in Nature Communication.

Thank you for your comments.

1. The authors stated that “The COFs properties including electronic conductivity, dipole moments, reductive ability, hydrophilicity were well tuned via precisely charging-modifying pyridine strategy.”. However, in conclusion, they also thought that the carbon atoms in the ionic imidazole act as active sites and contribute most for ORR. What is the main reason for the enhanced ORR activity?

We appreciate this comment.

The electronic conductivity, dipole moments, reductive ability, hydrophobicity, and binding ability of the O-contained intermediate were the properties of the COFs' bulks, and the carbon atoms in the ionic imidazole was the catalytic centers in the COFs. The higher electronic conductivity promoted the electronic transport from the electrodes to the catalytic sites, and the stronger dipole moments promoted to bind facilitate the absorption of O-containing intermediates, and hydrophobicity benefited the mass transport in the catalytic process. The synergetic roles of these properties and the highly active carbons contributed to the high activity, rather than single factors.

2. From TEM images and nitrogen adsorption/desorption isotherms, we can see that there are many defects in im-PY-BPY-COF. These defects give rise to the formation of amorphous part (seen from TEM) and the decrease of surface area (seen from nitrogen adsorption). In addition, these defects also will contribute to the ORR as well. The authors did not consider the effect of defect in im-PY-BPY-COF at all.

We appreciate this comment.

COFs are a privileged class of reticular materials, which can integrate their unique features in the design of advanced electrocatalysts with well-defined structure and active sites for electrochemical applications. Up to now, some COF materials have been reported as effective electrocatalysts for various electrocatalytic applications (*Science*

2015,349,1028; *J. Am. Chem. Soc.* **2018**, 140, 1116; *J. Am. Chem. Soc.* **2020**, 142, 8104; *Angew. Chem. Int. Ed.* **2022**, 61, e202209583). However, these works rarely focus on the impact of the defects on their electrocatalytic performance. In this work, from the PXRD patterns, TEM and BET results, it could be concluded that all COFs involved in our work show good crystallinities and high surface areas (*Nature synthesis*, **2022**, 1, 382-392).

Figure R1. (a) the PXRD profiles; and (b) the LSV curves for high crystallinity (black) and low crystallinity (red) of PY-BPY-COF.

To reveal the influence of the defects on the catalytic properties, we have synthesized a COF with low crystallinity as a control (*am*-PY-BPY-COF). And the PXRD patterns showed the *am*-PY-BPY-COF had a lower PXRD intensity, indicating it had more defects (Figure R1 a). We then investigate the catalytic performance (*am*-PY-BPY-COF). Accordingly, the *am*-PY-BPY-COF had the same E_0 and $E_{1/2}$ as those from the PY-BPY-COF (Figure R1 b). Therefore, although in general different defective sites and edge sites may have influence on the activity, in this work, the characterization data do not suggest defective is playing an important role in causing their catalytic activity difference for those three COF materials.

3. The importance of *im*-PY-BPY-COF should be emphasized with more structural characterizations and analyses during reaction instead of only using in situ SR-FTIR. Were there more defects after reaction? Did the COF decompose?

We appreciate this comment.

To further demonstrate the stability of COFs, the structure of the catalysts was

characterized by FT-IR and XRD. The FT-IR spectra revealed all the bonds were well retained, and the PXRD patterns showed the crystal structures were the same after long-term measurement. Thus, the decomposition of the COFs was ignored.

Figure S28. The XRD patterns of *im*-PY-BPY-COF before (black) and after (red) the chronoamperometry test.

Figure S29. The FT-IR spectra of *im*-PY-BPY-COF before (black) and after (red) the chronoamperometry test.

We have added sentences in the manuscript “The PXRD patterns and FT-IR spectra revealed that the crystal structure and frameworks were well retained after the long-term stability test, indicating high stability in 0.1 M solution (Supplementary Fig. 28 & 29).”

4. In Figure 3d and 3e, the authors cannot get the value of HOMO and LUMO by using UV-vis spectra solely. In addition, the band gap energy calculated from UV-vis spectra is not correct. Please check it again.

We appreciate this comment.

We have updated Fig 3e.

Fig. 3e Molecular orbitals and their energy diagrams calculated at the B3LYP/Def2-SVP level of PY-BPY-COF, *ion*-PY-BPY-COF, and *im*-PY-BPY-COF.

Fig. 3 a high-resolution N 1s spectra for PY-BPY-COF, *ion*-PY-BPY-COF, and *im*-PY-BPY-COF, respectively; b electronic conductivities; c EPR spectra under irradiation; d solid-state UV-vis absorption spectra of PY-BPY-COF (blue), *ion*-PY-BPY-COF (green) and *im*-PY-BPY-COF (red); e molecular orbitals and their energy diagrams calculated at the B3LYP/Def2-SVP level of PY-BPY-COF, *ion*-PY-BPY-COF, and *im*-PY-BPY-COF.

We have added sentences in the manuscript “To gain deeper insight into the electronic structures of PY-BPY-COF, *ion*-PY-BPY-COF, and *im*-PY-BPY-COF, density

functional theory calculations at the B3LYP/Def2-SVP level was performed. Those the synthesized three COFs displayed typical donor-acceptor configurations, in which the HOMOs (highest occupied molecular orbitals) are mainly located at the PY units of the COFs, while the (lowest unoccupied molecular orbitals) LUMOs are fully located over the linkers (Fig. 3e). The LUMO and HOMO energy-levels for PY-BPY-COF were -5.31 and -2.40 eV, with a corresponding energy gap of 2.91 eV. The *ion*-PY-BPY-COF possessed a lower HOMO (-8.75 eV) and substantially lower LUMO (-8.22 eV) energy and the *im*-PY-BPY-COF possessed a lower HOMO (-6.68 eV) and substantially lower LUMO (-5.48 eV) energy, indicating ions framework greater reductive ability.”

5. It is surprising that the ORR activity of *im*-PY-BPY-COF is so high. The authors have to check the ORR activity of molecular catalysts with the ionic imidazole structure as well.

We appreciate this comment.

We synthesized the structural models of three molecules (Supplementary Fig. 24) and tested their electrochemical properties by adding carbon nanotubes. Accordingly, the *im*-BPY achieved higher activity with an E_0 and $E_{1/2}$ than those from the BPY and *ion*-BPY.

Figure S24. ^1H NMR spectrum of (a) BPY (blue), (b) *ion*-BPY (green), and (c) *im*-BPY (red) recorded in $\text{DMSO-}D_6$ ($T=298\text{K}$, ppm).

Figure S25. The LSV curves of BPY (blue), *ion*-BPY (green), and *im*-BPY (red).

We have added sentences in the manuscript “To reveal the different catalytic activities of the linkers in the COFs, we have synthesized bipyridine (BPY), *ion*-bipyridine (*ion*-BPY) and *im*-bipyridine (*im*-BPY) (Supplementary Fig. 24), and investigated their catalytic activity (Supplementary Fig. 25). The LSV curves revealed that *im*-BPY had a higher activity with E_0 and $E_{1/2}$ of 0.88 and 0.72 V, which were positive than those from the BPY and *ion*-BPY. Thus, the catalytic activity for the small molecular was in the same trend with that from the COFs, and further confirming the contributions of the different linkers.”

6. The ORR activity of commercial catalyst (Pt/C) should not be so bad, please have a careful double-check and compare the activity of Pt/C to that from other literatures.

We appreciate this comment.

We have updated the catalytic performance of the commercial Pt/C catalysts and updated the manuscript.

We have added sentences in the manuscript “The onset potential (E_0) for Pt/C was 0.97 V, determined as the current density of 0.1 mA cm^{-2} , and the half-wave potential ($E_{1/2}$) was 0.85 V with a limited current density (j_{lim}) of 5.6 mA cm^{-2} (Fig. 4a, black curve).”

Fig. 4a LSV curves of Pt/C (black), PY-BPY-COF (blue), *ion*-PY-BPY-COF (green), and *im*-PY-BPY-COF (red).

7. There are many errors in this manuscript.

We appreciate this comment.

The language was improved by editing carefully and thoroughly.

8. What is the composition and morphology of the catalyst after electrochemical test?

We appreciate this comment.

To further demonstrate the stability of COFs, the structure of the catalysts was characterized by FT-IR and XRD. The FT-IR spectra revealed all the bonds were well retained, and the PXRD patterns showed the crystal structures were the same after long-term measurement. Thus, the decomposition of the COFs was ignored.

Figure S28. The XRD patterns of *im*-PY-BPY-COF before (black) and after (red) the chronoamperometry test.

Figure S29. The FT-IR spectra of *im*-PY-BPY-COF before (black) and after (red) the chronoamperometry test.

We have added sentences in the manuscript “The PXRD patterns and FT-IR spectra revealed that the crystal structure and frameworks were well retained after the long-term stability test, indicating high stability in 0.1 M solution (Supplementary Fig. 28 & 29).”

We appreciate this reviewer very much for the above valuable comments that greatly improved the quality of the manuscript.

REVIEWER COMMENTS

Reviewer #1 (Remarks to the Author):

Comments: After revisions, some of the issues were addressed. However, there were still some comments that were not addressed well.

1. Comment 1 of reviewer 1: FFT could be used to determine the structure of small area. I still think ED should be used to determine the COF structure in a large area.
2. Comment 4 of reviewer 1: The small molecules should be linked to the carbon nanotube, otherwise the charge transfer between carbon nanotube and small molecules could be a problem.
3. Comment 5 of reviewer 1: Author claimed that they have carefully checked the manuscript. However, even in the paragraph that author provided to prove they have revised the manuscript, there are many errors.
4. Comment 2 of reviewer 2: It is confusing. In Table S4, if author measured CHN element analysis, C, H, N of COF should be provided instead of only C + N.
5. Comment 4 of reviewer 2: The red line shows one reduction peak at 0.75 V and then go to higher current density. This line is abnormal.
6. Comment 2 of reviewer 3: I do not think the author understand the comment correctly. The current prepared COF contains abundant defects. But author prepared a COF with lower crystallinity to prove that the defects did not play an important role. It is not logically correct.
7. Comment 4 of reviewer 3: The HOMO and LUMO should not be only provided by DFT calculation. Some characterizations can be used to band gap and HOMO, such as solid UV, VB-XPS.

Reviewer #2 (Remarks to the Author):

The authors have attempted to address the concerns raised by the reviewers, and some aspects of the revised manuscript appear to have improved.

Reviewer #3 (Remarks to the Author):

In this manuscript, the authors constructed highly active catalytic centers via modulating the electronic states of the pyridine nitrogen atoms in the COFs for ORR. After modification, the quality of the article has been improved. Therefore, I recommend publication of this manuscript in Nat. Commun. after a minor revision.

1. In the revised manuscript, the authors listed some of the reported ORR activity of metal-free COFs. Please add this table in the main manuscript.
2. The authors demonstrated that “The high-resolution N 1s spectrum of PY-BPY-COF showed the peaks at 398.8 and 399.4 eV were from the pyridine and imine (Fig. 3a). With the ionization of the pyridine units, the peaks from N disappeared with two new peaks at 398.9 and 401.2 eV, which were from the pyridine radical cation and the positively charged N, confirming the existence of pyridinium in the ion-PY-BPY-COF.” Please cite related references.
3. The BET surface areas of these samples show great difference. Why? More related discussion should be provided.
4. The ORR activity of CNT should be added. During the measurement of catalytic activity, the introduction of CNT should be illustrated in the revised manuscript.

Reply to Reviewer#1:

After revisions, some of the issues were addressed. However, there were still some comments that were not addressed well.

Thanks for your encouragements and positive comments.

1. Comment 1 of reviewer1 (*Did the TEM image fit well with the simulated structure? Please provide ED pattern to confirm it.*): FFT could be used to determine the structure of small area. I still think ED should be used to determine the COF structure in a large area.

We appreciate this comment.

In the revision, the ED measurements were used to determine the COF structure in a large area. As shown in Supplementary Fig.7, all three COFs exhibited obvious diffraction patterns, which were assigned to the corresponding crystal facets (100), (220) and (330), respectively (*J. Am. Chem. Soc.* **143**, 7279-7284 (2021)). This is consistent with the FFT and PXRD results of the COFs, confirming that the crystalline structures were well maintained upon the modifications.

These results were provided in Supplementary Materials and discussed in the main text as “Additionally, the COFs were measured using selected area electron diffraction (SAED). The clear diffraction rings were observed from the SAED patterns, which were assigned to COF crystal planes of (100), (220), and (330), respectively (Supplementary Fig. 7). This is consistent with the FFT and PXRD observations of the COFs, confirming the well-maintained crystalline structures upon the modifications.”⁶⁶

Supplementary Fig.7 Selected area electron diffraction patterns of (A) PY-BPY-COF, (B) *ion*-PY-BPY-COF, and (C) *im*-PY-BPY-COF.

2. Comment 4 of reviewer1 (*...author could use molecules with such structure and link these molecules on carbon nanotubes*): The small molecules should be linked to the carbon nanotube, otherwise the charge transfer between carbon nanotube and small molecules could be a problem.

We appreciate this comment.

In the revision, the small molecules of [2,2'-bipyridine] -5,5'-dicarbaldehyde (BPY) were linked to the CNTs to obtain BPY-CNT, *ion*-BPY-CNT and *im*-BPY-CNT. The synthesis methods for these samples were described as follows:

Synthesis of BPY-CNT: 1 g of aminated CNT was mixed with BPY (50 mg) in dioxane solution (100 mL) at 80 °C for 2 h. The production was flitted, washed with THF, and ethanol, and then dried

under vacuum for 12 h to afford the BPY-CNT.

Synthesis of ion-BPY-CNT: BPY-CNT (250 mg) was dispersed in a mixture of acetonitrile (10 mL) and bromoethane (2.5 mL) in a flask. The mixture was degassed by the three freeze-pump-thaw cycles. The mixture was refluxed at 12 h under nitrogen protection. The production was filtered, washed with THF, and ethanol and dried under vacuum for 12 h to afford the *ion*-BPY-CNT.

Synthesis of *im*-BPY-CNT: In a flask, BPY-CNT (250 mg) and 0.3 mL of diiodomethane were dissolved in 10 mL of anhydrous acetonitrile under atmosphere. The reaction mixture was stirred at 100 °C for 48 h. The production was filtered, washed with THF, and ethanol and dried under vacuum for 12 h to afford the *im*-BPY-CNT.

In addition, the ORR performance on CNT, BPY-CNT, *ion*-BPY-CNT, and *im*-BPY-CNT samples were comparatively investigated. The LSV curves revealed that *im*-BPY-CNT had a higher activity with E_0 and $E_{1/2}$ of 0.86 and 0.74 V, respectively, which were more positive than those of aminated CNT (0.81, 0.66 V), BPY-CNT (0.82, 0.69 V), and *ion*-BPY-CNT (0.83, 0.70 V). Therefore, the BPY molecules and their ionized BPY molecules exhibited the similar trend in the terms of catalytic activity to that of COFs catalysts (Supplementary Fig. 29). This further confirmed that the ionized linkers contribute to the enhanced reactivity of *im*-PY-BPY-COF.

These results were provided in Supplementary Materials and discussed in the main text as “To eliminate the potential difference in charge transfer between carbon nanotubes (CNTs) and small molecules during the catalytic process, we linked small molecules to aminated CNTs. The LSV curves revealed that the catalytic activities of the small molecules attached to CNTs were similar to those of the COFs, further confirming the contributions of the different linkers (Supplementary Fig. 29).”

Supplementary Fig. 29 The LSV curves of aminated CNT (black), BPY-CNT (blue), *ion*-BPY-CNT (green), and *im*-BPY-CNT (red).

3. Comment 5 of reviewer1 (*There are many typos and errors in this manuscript and SI*): Author claimed that they have carefully checked the manuscript. However, even in the paragraph that author provided to prove they have revised the manuscript, there are many errors.

We appreciate this comment.

To improve the English language of this work, the manuscript has been revised carefully and thoroughly. We asked helps from a native English professor, and we also used the professional English editing service from Springer Nature (Figure Rev1).

Figure Rev.1 Proof of profession English editing services of Springer Nature.

4. Comment 2 of reviewer2 (*The elemental analysis of the synthesized compounds should be done with the more precise method, such as combustion analysis.*): It is confusing. In Table S4, if author measured CHN element analysis, C, H, N of COF should be provided instead of only C + N.

We appreciate this comment.

In the revision, the contents of elemental C, H, N were measured.

The results were provided in Supplementary Table 4 and discussed in the main text as “In addition, the elemental analysis results of C, N, and H were presented in Supplementary Table 4, showing agreement with the theoretical calculated values based on the networks”.

Supplementary Table 4. Elemental analysis of PY-BPY-COF, *ion*-PY-BPY-COF, and *im*-PY-BPY-COF.

Samples		C	N	H
PY-BPY-COF	Caclcd.	87.54%	7.56%	4.90%
	Found	85.11%	7.55%	4.62%
ion -PY-BPY-COF	Caclcd.	79.25%	6.49%	5.02%
	Found	78.27%	6.03%	5.23%

im -PY-BPY-COF	Cacl.	82.99%	6.91%	4.60%
	Found	81.32%	6.56%	4.18%

5. Comment 4 of reviewer 2 (...*The diffusion current density of im-PY-BPY-COF is abnormally high.*): The red line shows one reduction peak at 0.75 V and then go to higher current density. This line is abnormal.

We appreciate this comment.

In the revision, the ORR performance on *im*-PY-BPY-COF was re-tested (Figure R1). Correspondingly, the Tafel curves, TOF values, MA values, and normalized LSV curves on *im*-PY-BPY-COF were revised and updated in Figure 4.

These results were discussed in the main text as “Notably, *im*-PY-BPY-COF exhibited even better catalytic performances, with an E_0 of 0.90 V, $E_{1/2}$ of 0.80 V and j_{lim} of 5.8 mA cm^{-2} .”

Fig. R1 The retest LSV curve for the *im*-PY-BPY-COF in 0.1 M KOH.

Fig. 4 Electrochemical performance of COFs. a LSV curves; **b** corresponding Tafel slopes; **c** comparisons of the TOF and mass activities; **d** the C_{dl} ; **e** the normalized LSV curves by the ECSA; **f** electron transfer number and H_2O_2 yield plots determined from the RRDE measurements for Pt/C (black), PY-BPY-COF (blue), *ion*-PY-BPY-COF (green), and *im*-PY-BPY-COF (red).

6. Comment 2 of reviewer3 (From TEM images and nitrogen adsorption/desorption isotherms, we can see that there are many defects in *im*-PY-BPY-COF. These defects give rise to the formation of amorphous part (seen from TEM) and the decrease of surface area (seen from nitrogen adsorption). In addition, these defects also will contribute to the ORR as well. The authors did not consider the effect of defect in *im*-PY-BPY-COF at all.): I do not think the author understand the comment correctly. The current prepared COF contains abundant defects. But author prepared a COF with lower crystallinity to prove that the defects did not play an important role. It is not logically correct.

We appreciate this comment.

In fact, all three COFs have good crystallinity, which were supported by PXRD, HR-TEM and SAED measurements. In addition, we calculated the full width at half maximum (FWHM) of the diffraction peak to further confirm their crystal structures. The FWHM values of (100) were 1.07, 1.08, and 1.04 for PY-BPY-COF, *ion*-PY-BPY-COF, and *im*-PY-BPY-COF, respectively (Fig. R2), indicating that the three COFs have the similar crystallinity.

Therefore, it is reasonable that the COFs have similarly low defective degrees, which thus contribute to the similar impacts on the ORR catalytic performance.

Fig. R2 FWHM of the (100) peak in the PXR D patterns of (a) PY-BPY-COF, (b) *ion*-PY-BPY-COF, and (c) *im*-PY-BPY-COF.

7. Comment 4 of reviewer3 (In Figure 3d and 3e, the authors cannot get the value of HOMO and LUMO by using UV-vis spectra solely. In addition, the band gap energy calculated from UV-vis spectra is not correct. Please check it again.): The HOMO and LUMO should not be only provided by DFT calculation. Some characterizations can be used to band gap and HOMO, such as solid UV, VB-XPS.

We appreciate this comment.

In the revision, the band positions of the COFs were investigated using solid UV (Supplementary Fig 19) and VB-XPS (Supplementary Fig 20).

These results were provided in Supplementary Materials and discussed in the main text as “By analyzing the Tauc plots derived from the UV–vis absorption spectra, the optical bandgaps were calculated as 1.98, 1.91, and 1.76 eV for PY-BPY-COF, *ion*-PY-BPY-COF, and *im*-PY-BPY-COF, respectively (Supplementary Fig.19). The narrow bandgap of *im*-PY-BPY-COF favour electronic transport. Additionally, the valence band-X-ray photoelectron spectroscopy (VB-XPS) measurements of the valence band positions for PY-BPY-COF, *ion*-PY-BPY-COF, and *im*-PY-BPY-COF were yielded values of 1.33, 1.37, and 1.38 eV, respectively (Supplementary Fig.20).”

Supplementary Fig. 19 (A) The Tauc plots; valence band positions of the (B) PY-BPY-COF, (C) *ion*-PY-BPY-COF, and (D) *im*-PY-BPY-COF.

Supplementary Fig. 20 The band positions of the PY-BPY-COF, *ion*-PY-BPY-COF, and *im*-PY-BPY-COF.

We appreciate this reviewer very much for the above valuable comments that greatly improved the quality of the manuscript.

Reply to Reviewer#2:

The authors have attempted to address the concerns raised by the reviewers, and some aspects of the revised manuscript appear to have improved.

Many thanks for your strong support.

Reply to Reviewer#3:

In this manuscript, the authors constructed highly active catalytic centers via modulating the electronic states of the pyridine nitrogen atoms in the COFs for ORR. After modification, the quality of the article has been improved. Therefore, I recommend publication of this manuscript in Nat. Commun. after a minor revision.

Thanks for your encouragements and positive comments.

1. In the revised manuscript, the authors listed some of the reported ORR activity of metal-free COFs. Please add this table in the main manuscript.

We appreciate this comment.

The comparison of ORR activity of *im*-PY-BPY-COF and reported metal-free COFs was listed in Table 1 in the main manuscript.

As shown in Table 1, the *im*-PY-BPY-COF catalyst achieved higher activity than those reported COFs based catalysts.

Table 1. Comparison of the ORR activity of *im*-PY-BPY-COF with various recently reported metal-free COFs catalysts in 0.1 M KOH aqueous solution.

Catalysts	E _{1/2} (V vs. RHE)	E ₀ (V vs. RHE)	Tafel slope (mV dec ⁻¹)	TOF (s ⁻¹)	References
im -PY-BPY-COF	0.80	0.92	56.2	0.0170	This work
COF-JLU82	0.68	0.98	72.8	0.0044	70
Azo-COF	0.68	0.88	89.0	0.0025	55
JUC-528	0.70	0.83	65.9	0.0032	42
TAB-HKH-COF-CNT	0.79	0.86	42.7	0.0013	71
Oxazole -COF	0.75	0.85	79.0	0.0133	53
JUC-606	0.70	/	59.9	0.0016	72
JUC-616	0.78	1.02	52.9	0.0062	73
JUC-610-CON	0.72	0.83	61.9	0.0035	74
BTT-TAT-COF	0.77	0.87	71.0	0.0028	75
JUC-607	0.72	0.85	61.0	0.0022	76

2. The authors demonstrated that “The high-resolution N 1s spectrum of PY-BPY-COF showed the

peaks at 398.8 and 399.4 eV were from the pyridine and imine (Fig. 3a). With the ionization of the pyridine units, the peaks from N disappeared with two new peaks at 398.9 and 401.2 eV, which were from the pyridine radical cation and the positively charged N, confirming the existence of pyridinium in the ion-PY-BPY-COF.” Please cite related references.

We appreciate this comment.

The related references of *Angew. Chem. Int. Ed.* **62**, e202217527 (2023); *Nat. Commun.* **14**, 3800 (2023) and *Angew. Chem. Int. Ed.* **60**, 9642-9649 (2021) were cited as Ref. 63, Ref. 67 and Ref. 68, respectively, in the revision.

3. The BET surface areas of these samples show great difference. Why? More related discussion should be provided.

We appreciate this comment.

In the revision, more related discussion on the surface area differences of COFs were provided as “The decreased BET surface area and pore volume of *im*-PY-BPY-COF were attributed to stacking disorder caused by ionic repulsions, as well as pore filling by trifluoroacetate counterions. (*Nat. Synth.* **1**, 382-392 (2022)).”

4. The ORR activity of CNT should be added. During the measurement of catalytic activity, the introduction of CNT should be illustrated in the revised manuscript.

We appreciate this comment.

In the revision, the ORR performance of CNT was investigated. The E_0 , $E_{1/2}$ and the j_{lim} for CNT were 0.85 V (vs. RHE), 0.69 V, and 3.24 mA cm^{-2} , respectively (Supplementary Figure 30), which was much lower than that of *im*-PY-BPY-COF (E_0 of 0.90 V, $E_{1/2}$ of 0.80 V and j_{lim} of 5.8 mA cm^{-2}).

Supplementary Fig. 30 The LSV curve for the CNT in 0.1 M KOH.

These results were provided in the Supplementary Materials and discussed in the main text as “In addition, we investigated the ORR activity originating from the CNTs alone. The E_0 , $E_{1/2}$, and J_{lim} values for the CNTs were 0.85 V, 0.69 V and 3.24 mA cm^{-2} , respectively, indicating that observed activity was indeed originating from the COFs (Supplementary Fig. 30).”

We appreciate this reviewer very much for the above valuable comments that greatly improved the quality of the manuscript.

REVIEWERS' COMMENTS

Reviewer #1 (Remarks to the Author):

The authors have addressed my concerns in the revised version. It can be recommended for publication.

Reply to Referees' comments

Reply to Referee #1

Reviewer #1: The authors have addressed my concerns in the revised version. It can be recommended for publication.

=====

Thank you for your positive comments. We appreciate this reviewer very much for the above suggestive comments that greatly improved the quality of the manuscript.

=====